# Cryo-EM structure of a lysozyme-derived amyloid fibril from hereditary amyloidosis

Sara Karimi-Farsijani [1] ✉, Kartikay Sharma [1], Marijana Ugrina[2], Lukas Kuhn[1], Peter Benedikt Pfeiffer [1], Christian Haupt[1], Sebastian Wiese [3], Ute Hegenbart [4], Stefan O. Schönland [4], Nadine Schwierz [2], Matthias Schmidt [1] & Marcus Fändrich [1]

Systemic ALys amyloidosis is a debilitating protein misfolding disease that arises from the formation of amyloid fibrils from C-type lysozyme. We here present a 2.8 Å cryo-electron microscopy structure of an amyloid fibril, which was isolated from the abdominal fat tissue of a patient who expressed the D87G variant of human lysozyme. We find that the fibril possesses a stable core that is formed by all 130 residues of the fibril precursor protein. There are four disulfide bonds in each fibril protein that connect the same residues as in the globularly folded protein. As the conformation of lysozyme in the fibril is otherwise fundamentally different from native lysozyme, our data provide a structural rationale for the need of protein unfolding in the development of systemic ALys amyloidosis.

Systemic ALys amyloidosis is a disease that is based on the misfolding of human C-type lysozyme[1]. The disease affects individuals of different age groups, and patients show a wide range of symptoms from weight loss and gastro-intestinal bleeding, renal dysfunction, and sicca syndrome[2]. It is a hereditary disease that arises because of mutational changes in the lysozyme gene that encodes for lysozyme[3,4]. The disease is relatively rare even within the group of systemic amyloidosis. A study from 2020 reports less than 1 % of the patients with non-neuronal amyloidosis to be affected by systemic ALys amyloidosis[5]. The disease is monogenetic and depends invariably on mutational changes in lysozyme. Different lysozyme mutations have been described to underlie the disease in different patients, including the single-site mutational changes Y54N[6], I56T[3], F57I[7], W64R[8], D67G[9], D67H[3], L84S[10], D87G[2] and the double mutations T70N/W112R[11] and F3L/T70N[12]. Hence, the occurrence of any of these pathogenic mutations is small in the human population.

Lysozyme is an important component of the innate immune system that helps to neutralize pathogenic microbes. The protein is one of the best studied proteins in biochemistry and a textbook example of enzymatic function[13]. Natively folded lysozyme possesses a compact, globular structure and has been the archetype example of a protein

from the α + β class of protein folds. The protein contains two domains, an α-domain with mainly α-helical conformation and a β-domain containing mainly β-sheet structure[13]. Its native folding and misfolding characteristics have been studied extensively by several laboratories including the laboratory of the late Chris Dobson[14].

Biophysical analysis of the effect of disease-associated mutations revealed a mechanism, in which the native structure of lysozyme becomes destabilized by the mutations so that the mutant protein is prone to unfold into highly aggregation competent protein states[15,16]. While a native state-destabilizing activity of hereditary mutations has been well-documented in ALys and other forms of systemic amyloidosis[17,18], it is not known whether the mutations may contribute through further mechanisms to pathogenesis. For example, it may be possible that the mutations could modify the misfolding pathway and promote the formation of a specific fibril conformation. In absence of coordinate files showing the three-dimensional (3D) structures of the patient-derived ALys fibrils, it is difficult to further examine this possibility.

Here, we present the cryo-electron microscopy (cryo-EM) structure of an ALys amyloid fibril that was isolated from the amyloidotic tissue of a patient with systemic ALys amyloidosis. The patient suffered

[1]Institute of Protein Biochemistry, Ulm University, Ulm, Germany. [2]Institute of Physics, University of Augsburg, Augsburg, Germany. [3]Core Unit Mass Spectrometry and Proteomics, Medical Faculty, Ulm University, Ulm, Germany. [4]Medical Department V, Amyloidosis Center, Heidelberg University Hospital, Heidelberg, Germany. ✉e-mail: sara.karimi-farsijani@uni-ulm.de

from hereditary amyloidosis and expressed the LysD87G[2]. Analysis of the isolated fibrils with cryo-EM showed that they belong to a single fibril morphology and contain a single stack of fibril proteins. All residues of lysozyme contribute to formation of the stable fibril structure, which lack conformationally disordered fibril protein segments. The conformation of the fibril protein differs fundamentally from natively folded lysozyme, demonstrating the need of protein unfolding for the formation of the pathogenic fibril conformation.

## Results

### Isolation and characterization of LysD87G amyloid fibrils

ALys amyloid fibrils were isolated from the abdominal fat tissue of a previously described patient with systemic ALys amyloidosis arising from the D87G variant of human C-type lysozyme[2]. The patient was diagnosed at the age of 33 years and showed amyloid deposits in kidneys and abdominal fat tissue. Using a previously described isolation method[19], we isolated ALys fibrils from fat tissue to facilitate biophysical analysis. The fibrils were relatively pure, as indicated by a single protein band on a denaturing protein electrophoresis gel (Supplementary Fig. 1). The protein migrated at ~14 kDa, similar to previously described ALys amyloid fibril proteins[3]. Also, mass spectrometry shows that the fibrils consist mainly of full-length D87G lysozyme (Fig. Si 2). Transmission electron microscopy (TEM) showed well-resolved cross-overs in the fibril structure (Supplementary Fig. 3a). Scanning electron microscopy (SEM) and platinum side-shadowing additionally demonstrated the left-hand twist of the fibrils (Supplementary Fig. 3b).

### Reconstruction of the cryo-EM structure of the fibril

The fibrils were cryo-frozen and imaged with cryo-EM at 300 kV (Fig. 1a). There was an essentially monomorphic distribution of fibrils with no discernible polymorphism. The fibril width was determined at $106 \pm 9$ Å, while the fibril crossovers occurred at a regular distance of $584 \pm 58$ Å ($n = 30$; Supplementary Fig. 4). The 2D (Supplementary Fig. 5a) and 3D classes (Supplementary Fig. 5b) also showed no particle heterogeneity. The three-dimensional (3D) map was reconstructed at a final resolution of 2.8 Å, based on the 0.143 Fourier-shell correlation (FSC) criterion (Supplementary Fig. 6b). The atomic model, which we derived from the 3D map (Fig. 1a), depicts a polar fibril with C1 helical symmetry. The fibril consists of a single stack of fibril proteins; this is, the fibril contains a single protofilament (Fig. 1b). All 130 amino acid residues of native lysozyme could be fitted to the 3D map and formed

the fibril core (Fig. 1c). The absence of conformationally disordered segments in this fibril contrasts to the vast majority of fibril structures from systemic amyloidosis, which typically contain conformationally disordered segments, irrespective of whether the fibrils were purified from human or animal tissue[20–29].

### Fibril protein fold and enclosed structural cavities

The fibril protein shows an all-β fold that lacks elements of α-helical conformation (Fig. 2a,b). This conformation differs starkly from the structure of natively folded lysozyme, which contains elements of both α-helical and β-sheet structure (Fig. 2a,c). The fibril encompasses fifteen cross β-sheets (β1-β15) with strands varying from two to nine amino acid residues (Fig. 2a, b). The relative orientation of adjacent β-strands is uniformly parallel in the direction of the fibril main axis (Fig. 2b). There are three internal cavities that are termed here A, B and C (Supplementary Fig. 7a). Cavity A is surrounded by hydrophobic chemical groups and harbors a small density feature that could not be assigned to lysozyme (Supplementary Fig. 7a). This density feature is continuous in the direction of the fibril z-axis (Supplementary Fig. 7b) and could represent a non-polar molecular inclusion, such as a lipid, that runs across multiple layers of fibril proteins. Similar types of inclusions have been reported for several other ex vivo amyloid fibril structures derived for example from immunoglobulin light chain[21] or tau protein[30]. Cavities B and C are lined with polar amino acid residues and probably water-filled (Supplementary Fig. 7a). A computational evaluation of the fibril structure with the Amber function in Chimera[31] confirms this view and shows that the cavities are large enough to accommodate water molecules (Supplementary Fig. 8).

### Molecular interactions stabilizing the fibril structure

The fibril protein fold is stabilized by four intramolecular disulfide bonds that occur at residues Cys6-Cys128, Cys30-Cys116, Cys65-Cys81 and Cys77-Cys95 (Fig. 2b). These disulfides correspond to the four disulfide bonds in natively folded lysozyme (Fig. 2c). The disulfides impose severe, covalent constraints on the structure that help to prevent the unfolding of the fibril. There are a range of non-covalent interactions, such as buried clusters of hydrophobic residues - one such cluster is formed by residues Leu8, Ala9, Leu12, Leu15 and Val125 (Supplementary Fig. 9) - or buried pairs of residues with opposing charge, e.g. Asp67 and His78 (Supplementary Fig. 10a). Some non-covalent interactions run across different molecular layers of fibril proteins, such as the interaction between Arg41 from layer *i* and Asp53

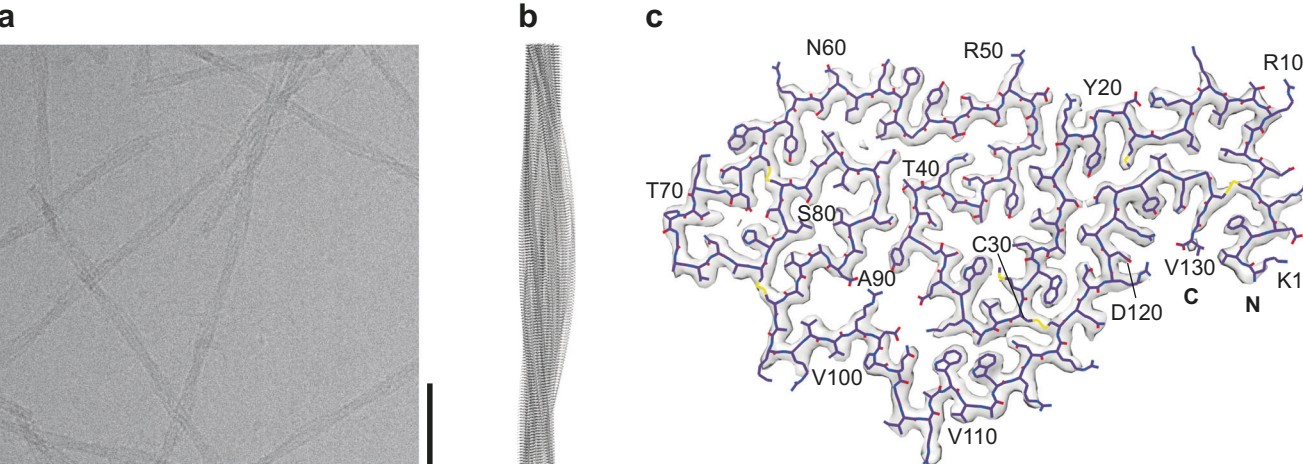

**Fig. 1 | Cryo-EM structure of an ex vivo ALys amyloid fibril. a** Representative cryo-EM image of a total of 2013 micrographs of the isolated fibrils ($n = 1$, biological sample). Scale bar: 50 nm. **b** Side-view of the reconstructed 3D map. **c** Cross-section of the 3D map (gray) overlaid with the atomic model (purple). The protein N and C-terminal ends as well as every tenth amino acid are labeled in the figure.

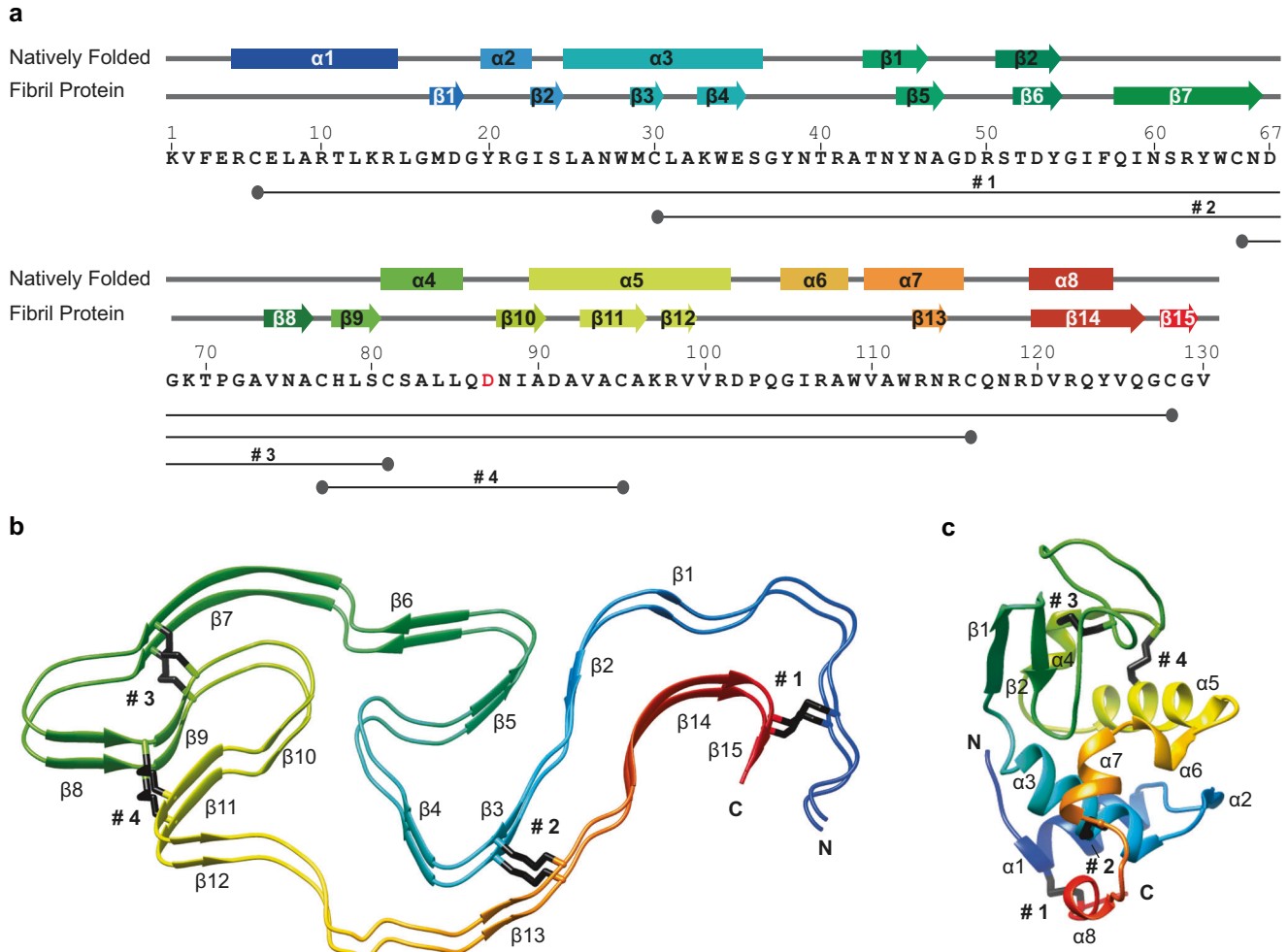

**Fig. 2 | Secondary structure of lysozyme fibrils and the natively folded protein. a** Sequence of human LysD87G. The mutant residue is marked in red. Above the sequence is a schematic representation of the secondary structural elements of the fibril protein and natively folded lysozyme. Arrows: β-sheets, boxes: α-helices. The four disulfide bonds (#1 to #4) are drawn as bars below the sequence. **b** Ribbon diagram of two molecular layers of the fibril. **c** Ribbon diagram of native human WT lysozyme (protein data bank: 1JSF[62]), scaled equally to the fibril protein. N- and C-terminal ends of the protein, secondary structural elements and disulfide bonds are labeled.

from layer $i − 1$ (Supplementary Fig. 10b). Cross-layer interactions occur because the fibril protein extends by 19 Å in the direction of the fibril z-axis, as measured as the distance between the highest and lowest Cα atom along the fibril z-axis (Supplementary Fig. 11a). Evaluation of the intermolecular interfaces of the fibril with the program PDBePISA[32] shows that each layer is in contact with nine other layers or fibril protein molecules (Supplementary Fig. 11b). The associated noncovalent interactions across the molecular layers of the fibril help to sterically interlock its architecture.

### Location of aggregation-prone segments in the fibril structure

Analysis of the fibril protein sequence with computational methods shows that the most aggregation-prone segments do not always match the segments that form the fibril cross-β structure (Supplementary Fig. 12a). This observation indicates that the intrinsic aggregation propensity of the protein sequence does not suffice to determine the observed fibril structure, although it may help to kinetically promote fibril formation or to form a certain fibril nucleus. The most aggregation-prone segments occur in the vicinity of the disulfide bonds Cys30-Cys116, Cys65-Cys81 and Cys77-Cys95. This association suggests that the three disulfide bonds occur at a site that may be important to trigger the aggregation process; and interestingly, many

ALys amyloidosis-associated mutations are also located within these segments (Supplementary Fig. 12b).

### Importance of the D87G mutation for the fibril

A previous analysis of the effect of the D87G mutation on the native protein conformation could not detect any strong destabilizing effect of this mutation on the globular lysozyme fold[2]. We thus analyzed the possible effect of this mutation on the fibril structure. The mutant residue (position 87) occurs within a tightly packed, hydrophobic β-arc of the fibril that is formed by residues Cys77-Cys95 (Supplementary Fig. 9). Residue 87 is a small glycine in the patient but a bulky and charged aspartate residue in wildtype (WT) lysozyme, suggesting that the WT protein is not compatible with the observed fibril structure. To test this possibility, we analyzed the experimentally determined fibril structure and a computationally generated homology model, which contains an aspartate residue at position 87 as in WT lysozyme, with different methods. Using programs such as Foldx[33], Amyloid Illustrator[34] and PDBePISA[32], there were some differences in the computed energy values of the patient fibril and the hypothetical WT counterpart (Fig. Si 13). However, the different programs varied considerably concerning the absolute height of the calculated stabilities, the stability contributions included in each program and the relative

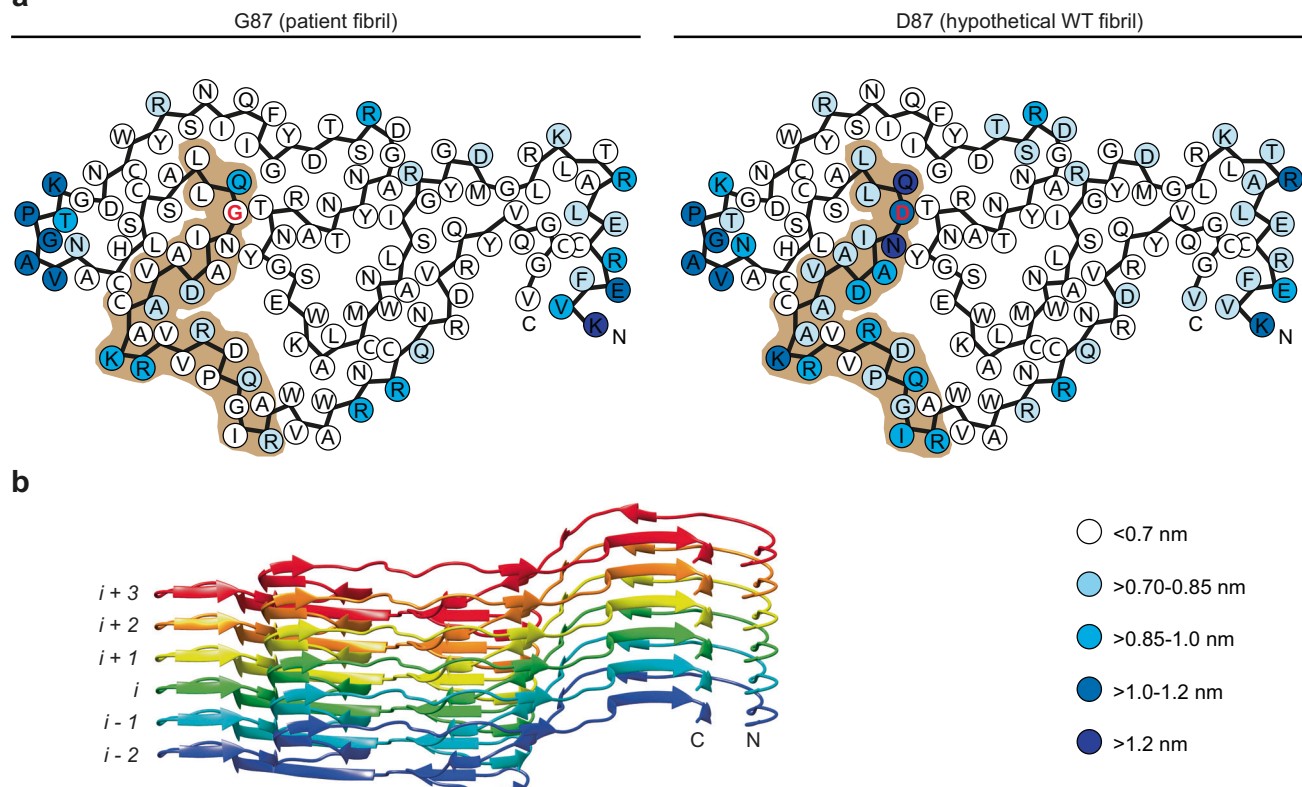

**Fig. 3 | The WT D87 residue leads to a local loosening of the fibril structure.**
**a** Schematic representation of the fibril protein with each residue being color-coded by the maximum distance between layers $i + 2$ and $i + 3$ adopted in the course of the MD simulation (blue scale and indicated in the panel). Brown: main region (Leu84 to Arg107) affected by the mutation. The figure was constructed based on the average values plotted in Fig. Si 13. Left: patient protein fibril with a glycine at position 87; right: hypothetical WT fibril structure, containing a D87 residue.
**b** Ribbon diagram of a six-layer stack of the patient fibril indicating the definition of layers $i − 2$ to $i + 3$.

stability order of the two fibril forms (Supplementary Fig. 13). There-fore, we were not able to discern any consistent trend whether or how the WT D87 residue influences the fibril structure.

Molecular dynamics (MD) simulations carried out with an explicit solvent model showed only small differences between the WT and the patient fibril. The most profound difference was seen for the molecules at one of the terminal ends of the fibril and in a segment of residues in the vicinity of the mutant position. This difference could be revealed by analyzing the distances between corresponding residues in suc-cessive layers of the fibril adopted in the course of the MD simulation. In a sequence segment extending from residue Leu84 to Arg107 of the WT fibril, we find many residues to show a distance of more than 0.7 nm to the corresponding residues in the adjacent fibril layer (Fig. 3). That is, the WT fibril shows a local loosening of the structure, which is not seen in the patient fibril, that contains a G87 residue, and in the remaining layers of the WT fibril (Supplementary Fig. 14).

**Location of disease-associated mutations in the fibril structure**
Analysis of the various known ALys amyloidosis-associated mutations does not indicate any obvious preference of the mutations for a spe-cific secondary structural element of the native protein or fibril structure (Supplementary Fig. 15a–c). Several mutations seem com-patible with the presently described fibril. Examples are the three mutations Y54N, F57I and W64R, which occur at solvent-exposed positions of the fibril (Supplementary Fig. 15c), or the relatively con-servative mutations F3L and T70N that affect buried positions of the fibril. Other mutations seem more difficult to reconcile with our fibril. Examples are the mutations I56T and L84S that occur at buried sites of the fibril where they surround the hydrophobic cavity A. Both

mutations replace a hydrophobic surface residue with a polar one (Supplementary Fig. 15c). The remaining three mutations (D67G, D67H and W112R) are buried and alter the chemical properties of the residue quite substantially. Plotted onto our fibril structure, they disrupt a buried ion pair or insert an uncompensated charge into a hydrophobic pocket. All of these mutations occur in the context of lysozyme molecules containing a D87 residue, which could be at least locally destabilizing to the fibril structure, as indicated above (Fig. 3, Sup-plementary Fig. 13,14).

## Discussion

Lysozyme is a vital component of the human innate immune system, although it may also form the basis of a debilitating protein misfolding disease. Using cryo-EM, we here obtained the structure of a fibril, that underlies the development of amyloidosis in one such patient. The fibril sample contained one dominant ALys fibril morphology that consisted of a single stack of fibril proteins. It was purified from abdominal fat tissue, which is an important amyloid deposition site within the human body that is even used for diagnostic purposes[5]. Previous studies established that the fibril morphology is consistent in fat tissue and other deposition sites within the human body[35]. Indeed, there is evidence from different amyloid diseases and from animals and humans that the fibril structure is consistent at different deposi-tion sites of the same patient or animal[29,35], all suggesting that the observed fibril structure is representative for the fibrils in this patient.

So far, it is not clear whether the observed fibril structure occurs also in ALys patients, who are associated by other lysozyme mutations. High-resolution structures of such fibrils are missing, and the available data provide a mixed picture. The absence of WT protein in the fibril

(Supplementary Fig. 2) suggests that lysozyme variants containing a D87 residue are not compatible with the fibril structure. Yet, there was no strong evidence from our theoretic examinations that a buried aspartate at position 87 cannot be tolerated by our fibril structure (Fig. 3, Supplementary Fig. 13, 14). Furthermore, several previous fibril structures show that charged residues that are not part of ion pairs can exist in amyloid fibril structures, such as the protein data bank (PDB) entries 8CG3[36], 7SAQ[37] and 6CU8[38].

Analysis of the nature of the lysozyme mutations in other ALys patients suggests that some are compatible with our fibril structure, while others may be unfavorable (Fig. Si 15). Some patients were reported to have fibril proteins that consist of full-length lysozyme[3], similar to the D87G lysozyme fibrils in our study. As the length of the fibril protein and its fragmentation is indicative of the fibril morphology[26,39], it is possible that the respective patients are affected by our fibril morphology as well. Still, other fibril structures than the ones reported here may be involved in systemic ALys amyloidosis, as other ALys patients have fibrils with fragmented lysozyme[11], suggesting that their fibrils are differently structured from ours.

The strong predominance of a single fibril morphology implies that this structure is particularly stable (biologically and/or thermodynamically) compared with other conformational states that may be formed from lysozyme. Indeed, several previous studies have suggested that pathogenic amyloid fibril structures are relatively stable in terms of their thermodynamic stability and/or resistance to proteolytic degradation[40,41]; and it is because of their biological stability that only specific amyloid fibril structures are able to accumulate and to cause damage inside the body. This selection of specific fibril morphologies because of their high proteolytic stability in vivo was termed 'proteolytic selection mechanism'[22].

The stability of the ALys fibril arises, on the one hand, from a number of non-covalent interactions that occur either within or across the molecular layers of the fibril (Fig. Si 10). On the other hand, there are four covalent cross-links provided by the protein's disulfide bonds that significantly constrain the fibril structure (Fig. 2b). These four disulfides connect the same eight residues that are also cross-linked in the natively folded structure of lysozyme (Fig. 2c), indicating that misfolding starts from a native protein conformation, which then misfolds into a specific, and pathogenic amyloid fibril state. The stark differences between the native protein and the fibril protein conformation demonstrate the need of unfolding of the native conformation for amyloid fibril formation. Our structural data thus explain the previous observation that the unfolding of lysozyme, in particular by mutational changes, is crucial to the development of systemic ALys amyloidosis[15,16].

The model for the transition of native lysozyme into a pathogenic amyloid fibril structure that emerges from our data emphasizes the need of unfolding for fibril formation. The predominance of a single fibril morphology in our samples further indicates the involvement of one or several mechanisms that prevent the formation or accumulation of other fibril morphologies inside the body. That is, it is either unfavorable for D87G lysozyme to form other fibril structures or it forms alternative fibril structures that are not stable in a biological environment, preventing them from being accumulated and becoming pathogenic. These conclusions are consistent with observations that amyloidosis is associated with fibril morphologies[41] of high proteolytic stability and the idea of pathogenic amyloid fibrils being proteolytically selected[22,41]. The great importance of the relative stabilities of the involved protein states further supports the idea that stabilizing the native state could represent a promising target to block the first step in the pathway and to ultimately combat the disease is humans. Indeed, this strategy has been employed with great success for the treatment of patients with systemic ATTR amyloidosis[42], a disease caused by the misfolding of transthyretin. Transthyretin is lipid-transporter protein of the serum that naturally binds the thyroid hormone thyroxine; and

the thyroxine binding site was used to design a complementary ligand that binds to the native protein and prevents its unfolding and further misfolding into amyloid fibrils. Since lysozyme also possesses a ligand binding cavity, that naturally binds bacterial peptidoglycans, it may be possible that this site may also be used to develop complementary ligands that stabilize the native protein fold and ultimately help to alleviate the problems associated with lysozyme misfolding inside the human body.

## Methods

### Ethical statement
The patient's written informed consent was obtained before the biopsy was taken. The collection and use of methods were approved by the ethics committees of the University of Heidelberg (123/2006) and the University of Ulm (203/18).

### Patient description
The fibrils used in this study were extracted from the subcutaneous abdominal fat tissue of a female patient in her 50 s suffering from systemic D87G ALys amyloidosis, confirmed by kidney biopsy. The abdominal fat tissue was removed from the patient with a 16-gauge needle and stored at −80 °C until fibril purification. The patient presented with a dominant renal involvement, but also with gastrointestinal symptoms and polyneuropathy[2]. The D87G mutation was also detected in the child of the patient, who is also suffering from amyloidosis. Later in life, the patient reached end-stage renal failure and was listed for a kidney transplantation.

### Fibril extraction from abdominal fat tissue
ALys amyloid fibrils were extracted from ~125 mg amyloid-laden abdominal fat tissue, following a previously described method with some modifications[19]. In brief, the tissue was placed in 0.25 mL Tris-(hydroxymethyl)-aminomethane (Tris) calcium-buffer (TCB), pH 8.0, which consisted of 20 mM Tris, 138 mM NaCl, 2 mM $CaCl_2$ and 0.1% (w/v) $NaN_3$. The sample was vortexed and spun down for 5 min at 3100 × $g$ and 4 °C. The washing step was repeated eight more times and the supernatants from each step were retained. We then dissolved one tablet of ethylenediamine tetra-acetic acid (EDTA)-free protease inhibitor (Roche) in 7 mL TCB and mixed 0.25 mL of this solution with 1.25 mg *Clostridium histolyticum* collagenase (Sigma-Aldrich) to obtain a final concentration of 5 mg/mL. The pellet from the last washing step was resuspended in 0.25 mL of the freshly prepared collagenase solution in TCB and incubated over night at 37 °C. In this incubation step the sample was constantly agitated at 700 rpm in an IKA MTS 2/4 digital table shaker. The sample was spun down for 30 min at 3100 × g and the top fat layer was removed. The pellet was washed three times in 125 μL Tris-EDTA-buffer (TEB, 20 mM Tris, 140 mM NaCl, 10 mM EDTA, 0.1 % (w/v) $NaN_3$, pH 8). The pellet obtained after the last TEB washing step was resuspended in 125 μL ice-cold water and centrifuged. The fibril-containing supernatant was retained and the water extraction was repeated eight more times. The supernatants from all water extraction steps were analyzed by gel electrophoresis or electron microscopy for the presence of fibrils. The fractions containing most fibrils were retained and used for further analysis.

### Denaturing protein gel electrophoresis
A 6 μL aliquot of the respective protein sample was mixed with 1.2 μL of 10 × NuPAGE reducing agent (Thermo Fisher Scientific), 3 μL of 4 × NuPAGE LDS sample buffer (Thermo Fisher Scientific) and 1.8 μL water. The mixture was heated at 95 °C for 10 min and loaded onto a 4–12% NuPAGE Bis-Tris gel (Thermo Fisher Scientific), which was run in NuPAGE MES LDS running buffer (Thermo Fisher Scientific) as recommended by the vendor. BlueEasy prestained protein ladder (Genetics) was used as a size marker. The gel was stained in a solution containing 30 % (v/v) ethanol, 10 % (v/v) acetic acid and 0.25 % (w/v)

Coomassie brilliant blue and destained with a solution containing 20 % (v/v) ethanol and 10 % (v/v) acetic acid.

### Mass analysis by mass spectrometry

A 100 µL aliquot of the fibril sample was lyophilized and dissolved in 4 µl 6 M guanidinium hydrochloride, 50 mM Tris, pH 8.0. After incubation at 4 °C over night, 11 µl of 0.1 % (v/v) trifluoroacetic acid (TFA) was added to the sample and analyzed by a U3000 RSLCnano high performance liquid chromatography (HPLC) system (Thermo Fisher Scientific). The sample was injected onto an Acclaim PepMap 100 precolumn (0.3 mm×5 mm) (Thermo Fisher Scientific) equilibrated with 0.1 % (v/v) TFA. In order to remove contaminants, the sample was washed with 0.1% (v/v) TFA for 5 minutes at a flow rate of 30 µL/min. Subsequently, the pre-column was switched in line with an Acclaim PepMap RSLC nanoViper column (0.075 × 500 mm) (Thermo Fisher Scientific). Proteins were eluted by applying a linear gradient from 5% to 40% HPLC solvent B (0.1% (v/v) formic acid, 86% (v/v) acetonitrile) in HPLC solvent A (0.1% (v/v) formic acid) over 30 min at a flow rate of 250 nL/min.

The HPLC was online coupled to an Orbitrap Elite mass spectrometer using the nano-Spray interface (both Thermo Scientific). For ionization, distal coated Silica-Tips (Bruker Daltonics) and the following parameter were used: spray voltage: 1.5 kV, capillary temperature: 250 °C; S-Lens radio frequency level: 68.9%. The mass spectra were recorded in the range from m/z 400 to m/z 1700 at a resolution of 30,000. The mass spectrometer was externally calibrated. The recorded mass spectra were summed over the chromatographic peak and deconvoluted with the FreeStyle 1.8 (ThermoScientific).

### Negative staining TEM

To prepare TEM negative samples, 3.5 µl of the sample was applied to a previously glow-discharged 200-mesh copper grid coated with formvar and carbon (Electron Microscopy Sciences). After incubating the sample at room temperature for 1 min, the excess solvent was carefully removed from the edge by filter paper. The copper grid was stained three times with a 2% (w/v) uranyl acetate and then washed three times with Millipore water. Afterwards, it was allowed to dry thoroughly and examined with a JEM-1400 transmission electron microscope (JEOL) at 120 kV.

### Platinum side shadowing

The handedness of the fibrils was determined by platinum side shadowing. A formvar and carbon-coated 200 mesh copper grid (Plano) was glow discharged for 30 s at 40 mA using a PELCO easiGlow glow-discharge cleaning system (Ted Pella). 1.5 µL fibril solution were applied on the grid and incubated for 30 min until it was dried up. Platinum particles were evaporated at an angle of 30° onto the grid to form a 1-nm-thick layer by use of a Balzers BAF 300 (Firma) coating device. Grids were imaged using a Hitachi S-5200 scanning electron microscope (Hitachi) provided with an acceleration voltage of 30 kV and operated with the secondary electron detector.

### Cryo-EM sample preparation and data collection

C-flat 1.2/1.3 400-mesh holey carbon-coated grids (Science Services) were glow-discharged at 40 mA for 30 s using a PELCO easiGlow glow-discharge cleaning system (Ted Pella). The fibril solution (i.e. supernatant fractions from the water extraction step) was diluted 1:3 with pure water and 3.5 µL of it was placed on the grid. The grid was plunge-frozen in liquid ethane after 8 s with a Leica G 2 (Thermo Fisher Scientific) plunging device using an incubation time of 30 s, >95 % humidity and back-side blotting with filter paper (Whatman). The grid was examined in at 200 kV in a JEM 2100 F (JEOL) transmission electron microscope to optimize the sample quality. The data set for cryo-EM reconstruction was recorded at 300 kV with a Titan Krios (Thermo Fisher Scientific) that was equipped with K2-Summit detector (Gatan). See Supplementary Table 1 for the data acquisition parameters.

### Reconstruction of the 3D map

The raw data given in tiff were converted into mrcs by IMOD[43]. The mrcs files were then imported into the software Relion 3.1.1[44], which was used for helical reconstruction. Motion correction was performed using MotionCor2[45]. For refining and correction of the contrast transfer function CTFFIND-4.1[46] was used. The particles were manually picked and extracted with a box-size of 300 Å and an interbox-distance of 33.5 Å, resulting into a total number of 120,356 particles. Afterwards a reference-free 2D classification was performed using 100 classes and a regularization parameter T = 2. No classes were dismissed and all particles were used for 3D classification. 3D classification consisted of one round using a cylinder as the initial model and the following other parameters: one class, T = 4, and a low-pass filter of 60 Å. The resulting 3D class was used as a reference model for the next rounds of classification, using 6 classes and T = 4. All classes seemed to belong to a common fibril morphology, and all contained particles were combined for further processing. The T value for following 3D classification with one class was increased to 20 and the result was used as improved reference model for the subsequent 3D auto-refinement and post-processing. This reconstruction was further improved using Bayesian polishing to achieve the final resolution of 2.8 Å, based on the value of the FSC curve for two independently refined half-maps at 0.143. An estimated map-sharpening B-factor factor of −75.9 Å$^2$. was applied by Relion post-processing. The final map had a twist of −1.4° and a helical rise of 4.8 Å. The reconstruction parameters are listed in Supplementary Table 1.

### Model building

The reconstructed 3D map was used for de-novo model building using Coot v0.9.8[47] software. In a first step, a poly-L-alanine chain was generated and threaded into one molecular layer of the fibril. The poly-L-alanine sequence was replaced with the lysozyme sequence of the patient protein. This model was validated in Phenix version v1.20.1[48] and further refined manually in Coot v0.9.8[47] with the aim to reduce the Ramachandran outliers, geometry outliers, rotamer outliers, and the clash score. After refining one molecular layer of the fibril, a fibril stack of nine layers was generated and refined. Multiplication of the layers was done using the pdbsymm tool from Situs[49]. The modeling parameters are listed in Supplementary Table 2.

### Aggregation score

Aggregation scores were calculated using TANGO[50], WALTZ[51], FoldAmyloid[52], Aggrescan[53], and PASTA2.0[54]. These programs reveal aggregation-prone segments. For this purpose, a value is calculated for each amino acid residue. The following settings were chosen for each program: WALTZ: threshold was set to best overall performance at pH of 7 and values above 75% are considered as hits; TANGO: Beta sheet aggregation values above 0.1 were considered as hits; FoldAmyloid: the scale was set to triple hybrid and amino acid residues having values above 0.062 for five consecutive residues are considered as hits and PASTA 2.0: region detection was set on 90% specificity and values below −2.8 PASTA Energy Units are considered as hits. An aggregation value of 0 means that none of the programs predict a high aggregation value for a particular amino acid residue, and conversely, an aggregation value of 5 means that all five programs predict a high aggregation value for that residue. The values from 0 to 5 were each coded with a color (Fig. Si 12), and the residues in the model were colored according to the value achieved by these 5 programs.

### Fibril stability calculations

The stabilities of the patient fibril with a G87 residue and the hypothetical WT fibril containing a D87 side chain, were calculated with

different computational tools. The Gibbs free energy of one molecular layer of the structures was calculated with the Amyloid Illustrator server[34]. Three-layer stacks of both fibrils were uploaded to the server and the free energy of one layer was obtained using the standard settings. Next, the energy difference between the unfolded and the folded state of the fibril proteins was calculated with FoldX, using the "stability of object" feature[33]. The obtained energy values for one molecular layer of the fibrils were inverted to reflect the energy gain upon the formation of the folded state. Finally, dissociation energies between two molecular layers of the fibrils along the z-axis were calculated with the program PDBePISA[32].The obtained values for the energy needed to dissociate two adjacent layers were inverted to obtain the energy gain upon the association of two layers.

## MD simulations

All-atom MD simulations were performed of the experimentally resolved lysozyme fibril structure and a homology model of this fibril, in which the sequence was reverted back to WT. To obtain the homology model, Gly87 was replaced by Asp using the PDB Manipulator tool of the CHARMM-GUI web-interface[55]. The algorithm replaced G87 with D87 such that the backbone dihedrals remained unchanged and the $NH_2$ atoms, which are identical in both residues, were placed at the same positions. The fibrils consisted of 6 protein layers, which formed a stable cross-β structure as in previous work[56]. The experimental structure and the homology model were placed in cubic simulation boxes with edge lengths of 146.5 Å and simulated at a temperature of 300 K. The boxes were filled with water and 0.15 M NaCl, leading to a total system size of about 412,950 atoms per box. The force field parameters for the proteins were taken from the Amber99sb-star-ildn force field[57] and the TIP4P-Ew[58] model was used for water. For NaCl, we used the Mamatkulov-Schwierz force field parameters[59].

The MD simulations were performed using the Gromacs simulation package, version 2020.6[60]. To prepare the starting structures, we used the steepest descent algorithm for energy minimization. This method iteratively adjusts atomic positions to lower the potential energy of the system, relaxing the structure to a local minimum. Convergence criteria were set to forces below 1000 kJ/mol/nm and convergence was achieved within 50,000 steps. Subsequently, we performed simulations at 300 K with a constant number of particles N, pressure P and temperature T. Periodic boundary conditions were applied, and the particle-mesh Ewald method was used for the periodic treatment of Coulombic interactions. Bonds to hydrogen atoms were constrained using the LINCS algorithm and a 2 fs time step was used. The systems were equilibrated for 1 ns in the NVT ensemble and then in the NPT ensemble using the Berendsen thermostat and barostat. For the production run, we performed 200 ns long simulations employing the velocity-rescaling thermostat with a stochastic term and a time constant $\tau_T = 0.1$ ps and isotropic Parrinello–Rahman pressure coupling with a time constant of $\tau_P = 5$ ps. To obtain sufficient statistics, each production run was repeated six times with the starting velocities drawn randomly from a Maxwell-Boltzmann velocity distribution. In total 2.4 μs of simulation time was used.

To quantify the dissociation of terminal chains, we calculated the maximum distance between two adjacent layers at the fibril tips over the course of the 200 ns simulations. The calculations were performed as follows: First, time dependent distances between equivalent amino acids in the two neighboring chains were calculated. Subsequently, maximum distances and standard error of the mean were calculated from the time dependent data. The first 50 ns of each trajectory were skipped in the calculations.

## Statistics

Whenever seen as appropriate the mean value and corresponding standard deviation was reported in this paper.

## Imaging and Morphology analysis

The images of the 3D map and protein model were prepared with the software UCSF Chimera v1.16[31]. Measurements of fibril width and crossover distance was done with Fiji[61].

## Reporting summary

Further information on research design is available in the Nature Portfolio Reporting Summary linked to this article.

## Data availability

The reconstructed 3D map was deposited in the Electron Microscopy Data Bank with the accession code EMD-18883. The coordinates files of the corresponding atomic model were deposited in the Protein Data Bank (https://www.rcsb.org/) under the accession code 8R4A. The following published PDB structure was used in the paper: PDB 1JSF. The cryo-EM raw data of the lysozyme derived fibrils were deposited on Electron Microscopy Public Image Archive (https://www.ebi.ac.uk/empiar/) with the accession code EMPIAR-11785. Mass spectrometry data was deposited at MassIVE [https://massive.ucsd.edu/ProteoSAFe/dataset.jsp?task=bc5cb0cfe8344f1494820fe9dcec289d], a member of the ProteomXchange consortium, under the accession code MSV000096142. Source data are provided with this paper.

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

## Acknowledgements
The authors thank Paul Walther (Ulm University) for technical support. Funding was partially provided by the Deutsche Forschungsgemeinschaft (FA 456/28 to M.F.). The collection of the cryo-EM datasets was funded by iNEXT Discovery (18586; Horizon 2020, European Union) and was done in the European Molecular Biology Laboratory in Heidelberg (Germany).

## Author contributions
S.K., K.S., L.K., M.U., N.S., P.B.P., C.H., S.W. and M.S. carried out research. S.K., K.S., M.U., N.S., M.S., C.H., S.W. and M.F. analyzed data. U.H., and S.O.S. contributed materials. M.F. designed research. S.K., M.S. and M.F. wrote the paper with contributions from all other authors.

## Funding

## Competing interests
The authors declare no competing interests.
