## [Peer Review file · Nature Communications]

Cryo-EM structure of a lysozyme-derived amyloid fibril from hereditary amyloidosis

Corresponding Author: Ms Sara Karimi-Farsijani

Version 0:

Reviewer comments:

Reviewer #1

(Remarks to the Author)

The research article titled "Cryo-EM structure of a lysozyme-derived amyloid fibril from hereditary amyloidosis." by Karimi-Farsijani et al describes the structure of the ex vivo cryo-EM fibrils from a systemic ALys amyloidosis patient with D87G mutation. This structure is the first ever determined from this type of amyloidosis, and is well resolved. Although this study may be interesting to the amyloidosis community, the following comments need to be addressed.

1. On page 5 results, under the heading Reconstruction of the cryo-EM structure of the fibril, the authors state "There was an almost monomorphic distribution of fibrils with no discernible polymorphism". What do the authors mean by "almost"? Do they find other species? Please elaborate and clarify whether they found a minor population with different conformations or can totally rule out the possibility of polymorphism.
2. I advise the authors to avoid using words such as "correlation" or "crosslink" for concepts that are not related to their most common definitions used in scientific writing, which can be misleading. Here are two examples.
 - a. On page 6, under the heading "location of aggregation-prone segments in the fibrils structure", the authors state, "Analysis of the fibril protein sequence with computational methods shows that the most aggregation-prone segments of the sequence do not correlate well with the fibril cross- β structure". "correlation" should be revised since the authors do not perform any correlation analysis.
 - b. On page 6, under the heading "Molecular interactions stabilizing the fibril structure", the authors state, "Altogether we find nine molecular layers of the fibril to be non-covalently cross-linked through this axial extension ". Please elaborate on the use of the word "crosslink" in this context. What do the authors mean with this? Additionally, please clarify in more detail the nature of these interactions. It would be quite useful to have a figure to show them.
3. Under the heading "Location of disease-associated mutations in the fibril structure", the authors assume that the mutations (Y54N, F57I and W64R) would not affect the fibril fold. At the same time, they say that the wild-type aspartic acid at position 87 is incompatible with the fibril fold. From a physiological standpoint, for the mutations(Y54N, F57I and W64R) to be present, the 87th position would comprise of the wild-type aspartic acid instead of glycine (since these mutations are point substitution), rendering the fibril fold incompatible, based on the authors study. These statements contradict each other. The familial mutations are not compatible with the presented fibril fold because of the aspartic acid 87. However, I can anticipate that, while I agree that D87G would play an impactful role in forming this arc, the other familial mutations may result in different fibril folds. Please revise these statements because they suggest that these other familial mutations could result in a similar fibril fold as in D87G, which may not be accurate. Also, please discuss the possibility of adopting different conformations that may be mutation specific.
4. The authors perform MD simulations to study the importance of D87G in the fibril fold stability. I think MD simulation is unnecessary in this situation. First, WT lysozyme has not been found in fibrils. Second, given the density map at that position, the introduction of a bulkier residue will definitely affect the local structure at the site and therefore, would be incompatible with the structure. I do not think this information is relevant.
5. In the discussion section, the authors state, "The strong predominance of a single fibril morphology implies that this structure is particularly stable (biologically and/or thermodynamically) compared with other fibril states that may be formed from lysozyme". This statement is not true. There are many amyloid structures found as unique structures in their

corresponding diseases, and yet, they are not particularly more stable than others that adopt multiple conformations (see all values in the Amyloid Atlas, Sawaya). And moreover, does that mean all fibrils from lysozyme will only have this structure regardless of the mutations? This is unlikely as discussed in comment #3. To make this statement, I suggest the authors to run a solvation energy calculation (or a hydrophobicity surface calculation), proteinase digestion profiling (comparing to polymorphic fibrils from other diseases), and/or stability assays (with SDS, urea, or other denaturing conditions, also compared to other fibrils that are polymorphic). These can reveal more information about the estimated stability of the fibrils that could be (only then) used to discuss the effect of this unique conformation in stability.

6. In the discussion on page 9, the authors state, "These four disulfides connect the same eight residues that are also cross-linked in the natively folded structure of lysozyme (Fig. 2c), indicating that misfolding starts from the native protein state,". It is really unclear what the authors want to convey here. Is there any example of amyloid fibrils that do not start with the misfolding of the native protein? Please revise.

7. On page 9, the authors say, "they can help to stabilize a specific structural conformer in the fibril state and thus help to shape up the pathogenic agent. This effect of the mutation is suggested both from a comparison of the fibril structure with the known mutational sites in the disease (Fig. 2c) as well as from an analysis of the fibril structure with MD simulations (Fig. 4)" The data do not show that the D87G mutation stabilizes the fibril structure that the authors describe. Instead, it shows that the wild-type residue (D) would probably be incompatible with that particular conformation. I would refrain from using the word stability when referring to the lack of steric hindrance. In my opinion (and without any stability data), the glycine at position 87 may likely be leading to an increase in flexibility that allows the formation of the arc and the folding into this particular conformation, rather than any possible interaction to improve fibril stability. Please revise.

8. On pages 9-10, the authors state, "that could indicate the involvement of an internal and proteolytic cleavage site due to an altered conformation. Other different mutations in systemic ALys amyloidosis may be associated with different fibril structures. A similar case was recently reported for A β (1-42) peptide, which forms different amyloid fibril structures, depending on whether the fibrils consist of the WT peptide or the E22G mutational variant³". The effect of proteolytic cleavage on fibril structure depends on whether the enzymatic activity happens before or after fibril formation. If it happens before, it is possible to envision that the different precursor fragments can lead to various conformational polymorphs. However, if proteolysis happens after the fibril is already formed, the presence of fragments may not associate with polymorphism. A very good example of this is ATTR amyloid fibrils. Please revise.

9. The authors need to perform a mass spectrometry analysis to confirm the presence of uncleaved fragments.

10. On page 12, the authors say ". 120,356 particles were manually picked with a box size of 300 Å and an inter box distance of 33.5 Å". Do the authors mean picked or extracted? The stats table also show that the number of particles remained the same after extraction, 2D and 3D processing. This seems nearly impossible (this would only happen if all the picked/extracted particles had the same high resolution and the same quality). Is this a typo? If not, can you explain how this was accomplished?

11. The fibril structure may also vary depending on the organ from which it is extracted. Therefore, the structure (and composition) of fibrils from this patient's kidney may be different from the fibrils extracted from fat aspirates, shown in the present study. This point may be worth discussing.

12. The authors should propose a model for the transition of the native structure to the fibril structure in the context of these results.

Minor concerns:

1. The authors are advised to mark with an arrow to indicate the crossover in sup fig 1.a.
2. On page 5, under the first subheading for results, the authors state, " and platinum side-shadowing additionally demonstrated the left-hand supertwist of the fibrils (Fig. S1 1b)." what do the authors mean by supertwist? Please elaborate.
3. Discussing the absence of wt amyloid deposits in vivo would be useful.
4. In supplementary figure 8, legend, please replace 19nm with 19Å.
5. In supplementary table 1, please replace "titon" krios with "titan" krios.
6. The authors need to show their 2D class averages as part of a supplementary figure
7. The authors need to show their 3D classification as well.
8. In Fig 2C the authors are advised to mark the position of the mutation in the figure.
9. It may be useful to briefly discuss the prevalence of this disease and the specific mutation D87G in the population.

Reviewer #2

(Remarks to the Author)

This is an additional cryo-EM study on amyloid fibrils from this group, this time showing the structure of fibril derived from human lysozyme with a D87G mutation. ALys amyloidosis is much studied but the structure of in vivo fibrils has not been described. The most important results are that the fibrils consist of full-length but more or less completely defolded and refolded lysozyme although the native disulfide bridges have been preserved, and that the studied mutation not only destabilizes the native structure of the protein but also stabilizes the fibril. The work is solid and the results are of great

interest.

It is a very nice study and I don't find any major problems with the manuscript but have some questions and minor remarks:

1. Reconstruction of the fibril (p. 5). This is slightly unclear to me. Do you mean that the fibril consists of only one protofilament?
2. The first paragraph of the discussion is mainly a repetition of the results and may be omitted.
3. Discussion, paragraph 3, last sentence: If the D87G mutation does not destabilize the native conformation, what would be the mechanism then?
4. Methods, fibril extraction. Why is collagenase necessary to extract the fibrils? Isn't there a risk that this process changes interactions, important for the fibril structure? Have you performed any control studies?

Version 1:

Reviewer comments:

Reviewer #1

(Remarks to the Author)

The authors addressed only some of my concerns. The authors did not address the following comments, which are (in my opinion) easy to address, and thus, raise further concerns.

1. The authors changed "almost" for "essentially" when talking about morphologies. This change is equally vague. Do they see other morphologies? If yes, they should be disclosed.
2. The mutation D87G allows for the folding of the protein into the fibril fold. It is not about stability. The authors fail to see the difference between these two and that deeply concerns me. The MD studies do not show that G87 is more stable than the wild-type D87. What shows is that D87 can not be accommodated into the mutant fold, and that is why the wild-type proteins dissociate from the structure on MD simulations. Similarly, the fitting of other disease-related mutations to the structure is not meaningful, since those other mutations will never be combined with the D87G mutation that would allow the formation of this particular structure. The structure of those disease-related mutations will surely be different, because the D87 residue in their sequence will not allow the formation of this particular structure. I do not see the point of having an entire section to say that the mutations are compatible but they would not form the structure because of D87. I recommend removing or at least reducing it significantly.
3. The MD simulations do not add anything to the paper. The wild-type is incompatible with the fibril fold, regardless of MD results. A residue replacement on Pymol or Chimera would show that the wild-type residue does not fit. I do not deem this experiment necessary.
4. To show that the residue G87 is adding to the stability of the fibril, multiple in silico tools can help, and there is no need to perform experiments in the wet lab. In silico tools include solvation energies shown multiple times for many cryo-EM structures (see Amyloid Atlas, by Sawaya), or those developed by Nikos Louros et al. I highly recommend performing these estimations if the authors want to be rigorous when talking about fibril stability.
5. The observation of one polymorph by cryoEM does not mean it is the only one or the most stable possible. The statement is still incorrect. There are many amyloid structures found as unique structures in their corresponding diseases, and yet, they are not particularly more stable than the same protein forming other structures. Please change.
6. Protein aggregation starts by definition from the native state (whatever that is, unfolded, folded, semi-folded...). What is the alternative? where is the misfolding start if not the native fold? I see it important to highlight that the disulfide bonds remain in the amyloid structure. That is an interesting observation perhaps suggesting that the unfolding needs to happen, there is no reduction during the process.
7. Did the author not discard any particles during 2D and 3D classification? This seems very unlikely (and not recommended). They also mentioned that "There was an essentially monomorphic distribution of fibrils with no discernible polymorphism", suggesting the presence of potential additional morphologies that were not discernible (therefore discarded?). Can the authors share all the 2D classes?
8. The sentence "there is evidence from different amyloid diseases and from animals and humans that the fibril structure is consistent at different deposition sites of the same patient or animal, all suggesting that the observed fibril structure is representative for the fibrils in this patient" is incorrect (it also has a typo and lack references). There are several studies (including some of the authors) on ATTR showing that patients with the same mutation (V30M) can show different structures depending on deposition site (vitreous humor vs heart). It is unclear whether the deposition does not vary the structure.
9. What do the authors mean by "These findings are consistent with observations in other forms of amyloidosis, where ex vivo fibrils were found to be structurally different and more proteolytically stable than in vitro formed fibrils, and hence with the idea that pathogenically relevant amyloid fibril structures arose from a proteolytic selection mechanism"?
10. What do the authors mean by "We conclude that all previously described fibril proteins may be difficult to reconcile with the currently described fibrils structure and that the fibrils in these patients could be structured differently"?

Reviewer #2

(Remarks to the Author)

The authors have satisfactorily addressed the issues raised by this reviewer

Reviewer #3

(Remarks to the Author)

The authors describe high resolution cryo-EM structure of patient-derived amyloid structure of mutant (D87G) lysozyme. The authors indicate that this is the first structure of such a fibril. Interestingly, the entire protein is well-resolved in the fibril state without the usual regions of unresolved disorder. As the first atomistically resolved structure of the class of lysozyme mutant fibrils, this is a noteworthy study. However, there are key points which I feel the authors should emphasize/expand and others in which the authors should de-emphasize/contract:

1. Is the disease due to loss of function of natively folded lysozyme or gain of function toxicity of the fibril form? The best consensus answer to this question is important to include because it determines the significance of the detailed structure of the fibril, which is the contribution of this work. If it is the former, then the shape of the fibril is only relevant so far as it informs on how much fibril can be formed (i.e. thermodynamic stability, kinetic accessibility, and proteolytic protection). If the answer is the latter, then knowledge of the shape of the fibril is important in its own right because of how it could inform on the toxicity mechanism. This is crucial context for the findings of this work.

2. As motivation for this work, the authors state that "In absence of detailed information about the three-dimensional (3D) structure of the formed fibrils it is difficult to further examine this possibility." I think this is too brief and indirect. The authors should instead state what types of structural characterization exists for lysozyme fibrils, and then state that lysozyme fibrils have not yet been characterized with atomic resolution. For example, kinetic and low-resolution structural characterization have been carried out by Ziaunys et al, Protein Science 2023, as well as by Matsuo et al, Frontiers in Molecular Biosciences, 2022.

3. Molecular dynamics simulations.

3a. It was never stated how many copies of the monomer were simulated. From Fig. 3b it looks like 5 or 6 monomer layers are included in each simulation. Is that correct? If so, the thinness of the layers would introduce bending and twisting modes of the systems that would not be present in experimentally observed fibrils. These deformations are incompatible with maintaining the in-register monomer-monomer stacking, leading to effective shear forces that could decouple the end-monomers. Even for the 87G mutant fibril, it looks like there are conformational fluctuations and decoupling of the end-monomers in Fig. 3b. Therefore, this simulation setup potentially introduces artifacts that artificially destabilize the fibril. I agree with the authors that the reversion of the mutant G87 to D is likely to destabilize the fibril due to negative charge repulsion and steric clashes, but a devil's advocate could say that the unfolding observed in the reverted simulations in Fig.3b is due to magnification of the destabilization by this simulation artifact. It could be argued that the D87 could in fact locally deform the fibril to make more space for the larger side chain without significantly altering/unfolding the fibril fold. The authors stated that they used a cubic box. In that case, this issue could be avoided by repeating the simulations in the same sized-box but with many more monomer copies (e.g. 20 or 30).

3b. In methods, authors stated that they used the PDB Manipulator tool to replace G87 by D and to position the D side chain. What were the constraints of this process? For example, was positioning done to optimize inter-molecular side-chain-packing with other monomers? Was the threshold for the following energy minimization? Details of this process should be provided. The starting point of the MD simulations for the homology-modeled WT fibril could be sensitive to the choices made in this process. This would then affect the stability of the system in the subsequent MD simulations. It is possible that there exists lower-energy alternative packings of the D87 that the MD does not sample in the relatively short time of 200 ns.

3c. It is standard practice to run 3 independent MD simulations at each condition to demonstrate that the results are robust to sampling variations arising from the stochastic nature of molecular dynamics.

Version 2:

Reviewer comments:

Reviewer #1

(Remarks to the Author)

The authors have addressed all concerns sufficiently.

Reviewer #3

(Remarks to the Author)

The authors adequately address my concerns.

Revision notes

Reviewer #1 (Remarks to the Author):

The research article titled "Cryo-EM structure of a lysozyme-derived amyloid fibril from hereditary amyloidosis." by Karimi-Farsijani et al describes the structure of the ex vivo cryo-EM fibrils from a systemic ALys amyloidosis patient with D87G mutation. This structure is the first ever determined from this type of amyloidosis, and is well resolved. Although this study may be interesting to the amyloidosis community, the following comments need to be addressed.

1. On page 5 results, under the heading Reconstruction of the cryo-EM structure of the fibril, the authors state "There was an almost monomorphic distribution of fibrils with no discernible polymorphism". What do the authors mean by "almost"? Do they find other species? Please elaborate and clarify whether they found a minor population with different conformations or can totally rule out the possibility of polymorphism.

Response: Thank you for critically evaluating our manuscript and for providing this generally favorable judgement. We could only recognize one fibril morphology with good confidence. However, since the sample contained also clumps of aggregated fibrils as well as short fibril fragment, we cannot totally exclude the presence of other structures. This is what we meant with the phrase "almost monomorphic". For better clarity, we now changed the word "almost" to "essentially", because we acknowledge the "almost" does somewhat conflict with the second half of the sentence.

2. I advise the authors to avoid using words such as "correlation" or "crosslink" for concepts that are not related to their most common definitions used in scientific writing, which can be misleading. Here are two examples.

a. On page 6, under the heading "location of aggregation-prone segments in the fibril structure", the authors state, "Analysis of the fibril protein sequence with computational methods shows that the most aggregation-prone segments of the sequence do not correlate well with the fibril cross- β structure". "correlation" should be revised since the authors do not perform any correlation analysis.

b. On page 6, under the heading "Molecular interactions stabilizing the fibril structure", the authors state, "Altogether we find nine molecular layers of the fibril to be non-covalently cross-linked through this axial extension ". Please elaborate on the use of the word "crosslink" in this context. What do the authors mean with this? Additionally, please clarify in more detail the nature of these interactions. It would be quite useful to have a figure to show them.

Response: To address the issues, we rewrote the text according to comment

a.) to now read as: "Analysis of the fibril protein sequence with computational methods shows that the most aggregation-prone segments do not always match the segments that form the fibril cross- β structure" and the text according to comment

b) to now read as: "Evaluation of the intermolecular interfaces of the fibril with the program PDBePISA shows that each layer is in contact with nine other layers or fibril protein molecules (Fig. Si 9b)."

We also revised the wording of section "Location of aggregation-prone segments in the fibril structure" to hopefully enhance its readability. However, the words correlation and cross-link were retained in the context of the Fourier-shell correlation criterion and the covalent cross-links provided by disulfide bonds.

3. Under the heading "Location of disease-associated mutations in the fibril structure", the authors assume that the mutations (Y54N, F57I and W64R) would not affect the fibril fold. At the same time, they say that the wild-type aspartic acid at position 87 is incompatible with the fibril fold. From a

physiological standpoint, for the mutations (Y54N, F57I and W64R) to be present, the 87th position would comprise of the wild-type aspartic acid instead of glycine (since these mutations are point substitution), rendering the fibril fold incompatible, based on the authors study. These statements contradict each other. The familial mutations are not compatible with the presented fibril fold because of the aspartic acid 87. However, I can anticipate that, while I agree that D87G would play an impactful role in forming this arc, the other familial mutations may result in different fibril folds. Please revise these statements because they suggest that these other familial mutations could result in a similar fibril fold as in D87G, which may not be accurate. Also, please discuss the possibility of adopting different conformations that may be mutation specific.

Response: Thank you for this comment. We think that there is no disagreement on the scientific content and the comment arises mainly from the description in the text, which so far dealt with the mutational site only, while the referee wants the mutations to be presented in the context of a full protein. In response to this comment, we changed the order of the paragraphs and sections in our results description and in the figures. Therefore, we now present the effect of the Asp87Gly mutation before considering other mutations and describing their effect in the context of a full protein. Through these changes we hope to have clarified that the mutated proteins are unfavorable for the observed fibril structure, although the mutations themselves may not contribute to this effect. In addition, we restructured the penultimate paragraph of the discussion to now discusses more clearly that fibril structures in ALys amyloidosis might be mutation-specific.

4. The authors perform MD simulations to study the importance of D87G in the fibril fold stability. I think MD simulation is unnecessary in this situation. First, WT lysozyme has not been found in fibrils. Second, given the density map at that position, the introduction of a bulkier residue will definitely affect the local structure at the site and therefore, would be incompatible with the structure. I do not think this information is relevant.

Response: Thank you for the comment. However, we find it generally more convincing to conduct an experiment than to solely trust our intuition. Therefore, we consider the obtained MD results as important.

5. In the discussion section, the authors state, "The strong predominance of a single fibril morphology implies that this structure is particularly stable (biologically and/or thermodynamically) compared with other fibril states that may be formed from lysozyme". This statement is not true. There are many amyloid structures found as unique structures in their corresponding diseases, and yet, they are not particularly more stable than others that adopt multiple conformations (see all values in the Amyloid Atlas, Sawaya). And moreover, does that mean all fibrils from lysozyme will only have this structure regardless of the mutations? This is unlikely as discussed in comment #3. To make this statement, I suggest the authors to run a solvation energy calculation (or a hydrophobicity surface calculation), proteinase digestion profiling (comparing to polymorphic fibrils from other diseases), and/or stability assays (with SDS, urea, or other denaturing conditions, also compared to other fibrils that are polymorphic). These can reveal more information about the estimated stability of the fibrils that could be (only then) used to discuss the effect of this unique conformation in stability.

Response: It is a common principle in physical chemistry that the observation of a single state implies that this state is more stable than possible alternative states. This principle underlies the Boltzmann distribution and Anfinsen's thermodynamic hypothesis, according to which the native state of a protein is the thermodynamically most stable (monomeric) state under native conditions. For this reason, we believe that it is generally fair to conclude: "the strong predominance of a single fibril morphology implies that this structure is particularly stable ... compared with other fibril states ...". Two additions to this statement are necessary, however: First, we mean "compared with other fibrils states that may be formed from lysozyme". The referee seems to have understood that we tried to compare lysozyme fibrils with from other proteins. Second, we do not necessarily mean thermodynamic stability but biological stability in terms of resistance to degradation by proteases,

macrophages etc. Indeed, we now conducted an additional experiment, which finds ALys fibrils to be similarly stable as previously described ex vivo AL amyloid fibrils (Figure R1), which are considerably more proteolytically stable than in vitro formed fibrils, such as in vitro fibrils from murine SAA protein (Bansal et al., 2021; Schönfelder et al. 2021). Taken together, the criticized sentence now reads as “The strong predominance of a single fibril morphology implies that this structure is particularly stable (biologically and/or thermodynamically) compared with other fibril states that may be formed from lysozyme.” We regret that a theoretic evaluation, such as the one suggested by the referee, cannot be performed as there are no alternative fibril morphologies of lysozyme reported so far.

Figure R1

Figure R1. Comparison of the proteolytic stability of amyloid fibrils using proteinase K. Two cases of ex vivo amyloid fibrils, LysD87G amyloid fibrils and AL amyloid fibrils, and one case in vitro, SAA fibrils, were analysed. The fibril-to-proteinase ratio is 1:10, and after the addition of proteinase K, the fibrils were incubated for different time steps and subsequently applied to the gel for analysis.

6. In the discussion on page 9, the authors state, "These four disulfides connect the same eight residues that are also cross-linked in the natively folded structure of lysozyme (Fig. 2c), indicating that misfolding starts from the native protein state,". It is really unclear what the authors want to convey here. Is there any example of amyloid fibrils that do not start with the misfolding of the native protein? Please revise.

Response: We feel that the referee agrees in full to our conclusion that fibril formation starts from a native lysozyme state. Instead, it is criticized that this conclusion is mentioned at all, because it sounds trivial to the referee. According to our experience, however, it is not generally accepted that misfolding always starts from a native state. We frequently get questions from other referees and from audiences at scientific presentations that challenge the view that misfolding starts from the native state. And indeed, no one was so far able, according to our knowledge, to trace the in vivo fate of single fibril precursor protein molecules from translation to fibril assembly. Due to this uncertainty we consider it as important to mention our conclusion about the relevance of the native state.

7. On page 9, the authors say, "they can help to stabilize a specific structural conformer in the fibril state and thus help to shape up the pathogenic agent. This effect of the mutation is suggested both from a comparison of the fibril structure with the known mutational sites in the disease (Fig. 2c) as well as from an analysis of the fibril structure with MD simulations (Fig. 4)" The data do not show that the D87G mutation stabilizes the fibril structure that the authors describe. Instead, it shows that the wild-type residue (D) would probably be incompatible with that particular conformation. I would

refrain from using the word stability when referring to the lack of steric hindrance. In my opinion (and without any stability data), the glycine at position 87 may likely be leading to an increase in flexibility that allows the formation of the arc and the folding into this particular conformation, rather than any possible interaction to improve fibril stability. Please revise.

Response: Our MD simulations show a (partial) dissociation of the fibril protein molecules at the terminal ends of the fibril, if the fibril protein contains the wildtype residue (Asp) at position 87. The patient fibril, which contains a Gly residue at this position, does not show this effect to the same extent. This means that the patient fibril is more stable. It retains its structure and does not dissociate as readily as the fibril structure containing an Asp.

8. On pages 9-10, the authors state, "that could indicate the involvement of an internal and proteolytic cleavage site due to an altered conformation. Other different mutations in systemic ALys amyloidosis may be associated with different fibril structures. A similar case was recently reported for A β (1-42) peptide, which forms different amyloid fibril structures, depending on whether the fibrils consist of the WT peptide or the E22G mutational variant³". The effect of proteolytic cleavage on fibril structure depends on whether the enzymatic activity happens before or after fibril formation. If it happens before, it is possible to envision that the different precursor fragments can lead to various conformational polymorphs. However, if proteolysis happens after the fibril is already formed, the presence of fragments may not associate with polymorphism. A very good example of this is ATTR amyloid fibrils. Please revise.

Response: We agree with the view that it is possible that patients with other mutations may involve other fibril structures. That was in fact supposed to be the point of the penultimate paragraph of the previous manuscript version. As this section was probably not clear enough, we revised it thoroughly.

9. The authors need to perform a mass spectrometry analysis to confirm the presence of uncleaved fragments.

Response: We have done a mass spectrometry (MS) analysis and report the data in the new Fig. S1. These data confirm what was already evident from gel electrophoresis and cryo-EM: fibrils contain full-length lysozyme.

10. On page 12, the authors say ". 120,356 particles were manually picked with a box size of 300 Å and an inter box distance of 33.5 Å". Do the authors mean picked or extracted? The stats table also show that the number of particles remained the same after extraction, 2D and 3D processing. This seems nearly impossible (this would only happen if all the picked/extracted particles had the same high resolution and the same quality). Is this a typo? If not, can you explain how this was accomplished?

Response: This sentence was rewritten to now read as: "The particles were manually picked and extracted with a box-size of 300 Å and an interbox-distance of 33.5 Å, resulting into a total number of 120,356 particles". That is, we did not discard any particles during 2D and 3D classification because there was no need for it; i.e. our classes did not suggest the involvement of significant levels of bad particles or of alternative fibril morphologies.

11. The fibril structure may also vary depending on the organ from which it is extracted. Therefore, the structure (and composition) of fibrils from this patient's kidney may be different from the fibrils extracted from fat aspirates, shown in the present study. This point may be worth discussing.

Response: Several studies do not support the conjecture that the fibril morphology varies with the tissue. We now discuss this issue in our Discussion: "The fibril was purified from abdominal fat tissue, which is an important amyloid deposition site within the human body that is even used for diagnostic purposes. The fibril was purified from abdominal fat tissue, which is an important amyloid deposition site within the human body that is even used for diagnostic purposes. Previous studies established that the fibril morphology is consistent in fat tissue and other deposition sites

within the human body. Indeed, there is evidence from different amyloid diseases and from animals and humans that the fibril structure is consistent at different deposition sites of the same patient or animal, all suggesting that the observed fibril structure is representative for the fibrils in this patient. “

12. The authors should propose a model for the transition of the native structure to the fibril structure in the context of these results.

Response: We added a possible model for the formation of fibrils as a text description at the beginning of the last paragraph: “The model for the transition of native lysozyme into a pathogenic amyloid fibril structure that emerges from our data emphasizes the need of unfolding for fibril formation. The predominance of a single fibril morphology in our samples further indicates the involvement of one or several mechanisms that prevent the formation or accumulation of other fibril morphologies inside the body. That is, it is either unfavorable for D87G lysozyme to form other fibril structures or it forms alternative fibril structures that are not stable in a biological environment, preventing them from being accumulated and becoming pathogenic. These findings are consistent with observations in other forms of amyloidosis, where ex vivo fibrils were found to be structurally different and more proteolytically stable than in vitro formed fibrils, and hence with the idea that pathogenically relevant amyloid fibril structures arose from a proteolytic selection mechanism”.

Minor concerns:

1. The authors are advised to mark with an arrow to indicate the crossover in sup fig 1.a.

Response: Thank you for your suggestion, we added yellow arrows to mark the cross-overs in Supplementary figure 1a.

2. On page 5, under the first subheading for results, the authors state, " and platinum side-shadowing additionally demonstrated the left-hand supertwist of the fibrils (Fig. Si 1b)." what do the authors mean by supertwist? Please elaborate.

Response: The supertwist refers to the twist of the entire fibril and not, e.g., to the beta-sheet twist. We now changed it to “twist”.

3. Discussing the absence of wt amyloid deposits in vivo would be useful.

So far, we had no information on the presence or absence of WT lysozyme in the fibrils but our new MS data (Fig. Si 1) show mainly the mutant protein in the fibrils. These data are further support to our view that WT lysozyme is incompatible with the observed fibril fold, which is now expressed in the section “Importance of the D87G mutation for the fibril”.

4. In supplementary figure 8, legend, please replace 19nm with 19Å.

Response: Thank you for spotting this error. We have changed it accordingly.

5. In supplementary table 1, please replace “titon” krios with “titan” krios.

Response: Thank you for spotting this mistake, we changed it accordingly.

6. The authors need to show their 2D class averages as part of a supplementary figure

Response: We added the 2D classes as supplementary figure 3a.

7. The authors need to show their 3D classification as well.

Response: We added the 3D classes as a supplementary figure 3b.

8. In Fig 2C the authors are advised to mark the position of the mutation in the figure.

Response: To show the position of the mutations in the fibril fold, in the native protein structure and in the sequence is already the purpose of Figure 3 (new Figure 4). As Figure 2 is already quite dense, we would prefer not to overload it.

9. It may be useful to briefly discuss the prevalence of this disease and the specific mutation D87G in the population.

Response: We now address this issue in the revised introduction: “The disease is relatively rare even within the group of systemic amyloidosis. A study from 2020 reports less than 1 % of the patients with non-neuronal amyloidosis to be affected by systemic ALys amyloidosis. The disease is monogenetic and depends invariably on mutational changes in lysozyme. Different lysozyme mutations have been described to underlie the disease in different patients, including the single-site mutational changes Y54N, I56T, F57I, W64R, D67G, D67H, L84S, D87G and the double mutations T70N/W112R and F3L/T70N. Hence, the occurrence of any of these pathogenic mutations is small in the human population.”

Reviewer #2 (Remarks to the Author):

This is an additional cryo-EM study on amyloid fibrils from this group, this time showing the structure of fibril derived from human lysozyme with a D87G mutation. ALys amyloidosis is much studied but the structure of in vivo fibrils has not been described. The most important results are that the fibrils consist of full-length but more or less completely defolded and refolded lysozyme although the native disulfide bridges have been preserved, and that the studied mutation not only destabilizes the native structure of the protein but also stabilizes the fibril. The work is solid and the results are of great interest. It is a very nice study and I don't find any major problems with the manuscript but have some questions and minor remarks:

1. Reconstruction of the fibril (p. 5). This is slightly unclear to me. Do you mean that the fibrils consists of only one protofilament?

Response: We thank this referee for these very constructive comments. Yes, we mean that the fibril consists of a single protofilament. We now reworded the text to read as: “The fibril consists of a single stack of fibril proteins; this is, the fibril contains a single protofilament.”

2. The first paragraph of the discussion is mainly a repetition of the results and may be omitted.

Response: Thank you for your suggestion. The discussion was significantly revised to account for the remarks of the other referee. The previous first paragraph is now distributed over two paragraphs that contain additional references to the literature and probably do not anymore give the impression of a summary of the results.

3. Discussion, paragraph 3, last sentence: If the D87G mutation does not destabilize the native conformation, what would be the mechanism then?

Response: Thank you for your question. We are not sure but it seems to us that there is a misunderstanding here. The last sentence of the previously third paragraph was actually meant to allow the possibility that the mutation destabilizes the native state. Hopefully, this has now become clearer, as we reworded this paragraph and the criticized sentence, which now reads as: “this fibril stabilizing effect of the D87G mutation does not rule out the possibility that the mutation additionally destabilizes the native protein conformation, similar to other amyloidogenic lysozyme mutations”.

4. Methods, fibril extraction. Why is collagenase necessary to extract the fibrils? Isn't there a risk that this process changes interactions, important for the fibril structure? Have you performed any control studies?

Response: Thank you for your question. The collagenase is needed to destroy the connective tissue around the fibrils. We have done (with other fibril proteins and in fat tissue) extensive tests without collagenase, but all alternative protocols never reached a convincing purity. That is, we were never sure whether protein bands on the gel represent integral fibril proteins and whether they were contaminants or peripherally attached protein components. Considering that we had only very little material for this study (~0.6 g of an abdominal fat aspirate), we had to rely on an established protocol and could not do further explorations with this case.

REVIEWER COMMENTS

Reviewer #1 (Remarks to the Author):

The authors addressed only some of my concerns. The authors did not address the following comments, which are (in my opinion) easy to address, and thus, raise further concerns.

1. The authors changed "almost" for "essentially" when talking about morphologies. This change is equally vague. Do they see other morphologies? If yes, they should be disclosed.

Response: We thank this referee for critically reading our manuscript and revisions. While we cannot rule out the existence of other fibril morphologies, we have no clear, positive evidence for their involvement. We occasionally saw fibrils of uncertain morphology that may or may not correspond to the main morphology. But even if these fibrils represented other fibril structures, they were rare. The sample is thus best described as “essentially monomorphic”.

2. The mutation D87G allows for the folding of the protein into the fibril fold. It is not about stability. The authors fail to see the difference between these two and that deeply concerns me. The MD studies do not show that G87 is more stable than the wild-type D87. What shows is that D87 can not be accommodated into the mutant fold, and that is why the wild-type proteins dissociate from the structure on MD simulations. Similarly, the fitting of other disease-related mutations to the structure is not meaningful, since those other mutations will never be combined with the D87G mutation that would allow the formation of this particular structure. The structure of those disease-related mutations will surely be different, because the D87 residue in their sequence will not allow the formation of this particular structure. I do not see the point of having an entire section to say that the mutations are compatible but they would not form the structure because of D87. I recommend removing or at least reducing it significantly.

Response: The analyses requested under comment 4 (see below) as well as by referee 3 regarding MD simulations provided new results that challenged (or did not clearly support) the idea that the wildtype D87 residue prevents the formation of the fibril structure. There were differences between Amyloid Illustrator, Foldx and PDBePISA regarding the absolute energy values and the relative stability order of the two fibrils (patient and wildtype), so that we could not discern a consistent trend which fibril structure is more stable (SI Fig. 12). MD indicates at least a local, D87-dependent destabilization of the structure within one terminal layer of the fibril (SI Fig. 13), while the remaining fibril layers remain almost unaffected. Since we previously assumed - similar to the referee - that an uncompensated internal charge of D87 is unfavourable, these new data are surprising. They led us to update the manuscript's conclusions regarding the effect of D87. Based on all available data, there is no clear evidence that the WT D87 residue is destabilizing; and in fact, buried and apparently uncompensated charges have been found in other amyloid fibril structures as well (e.g. PDB entries 8cg3, 7saq or 6cu8). We have revised the description of the other mutations in the results and discussion and moved the corresponding figure to the SI (SI Fig. 14). We hope that this could account for the referee's request to reduce the emphasis on this topic

3. The MD simulations do not add anything to the paper. The wild-type is incompatible with the fibril fold, regardless of MD results. A residue replacement on Pymol or Chimera would show that the wild-type residue does not fit. I do not deem this experiment necessary.

Response: Visualization of the mutational changes in the fibril structure with Pymol or Chimera can suggest an effect of the D87 residue, but does not suffice to demonstrate it. In the present case, visual inspection would have suggested a strong destabilizing effect that cannot be confirmed clearly by more direct experiments. A further advantage of MD simulations is that this can provide residue- and layer-specific structural information and reveal even subtle effects, as shown in SI Fig 13.

4. To show that the residue G87 is adding to the stability of the fibril, multiple in silico tools can help, and there is no need to perform experiments in the wet lab. In silico tools include solvation energies shown multiple times for many cryo-EM structures (see Amyloid Atlas, by Sawaya), or those developed by Nikos Louras et al. I highly recommend performing these estimations if the authors want to be rigorous when talking about fibril stability.

Response: We have now performed such calculations with Amyloid Illustrator, Foldx and PDBePISA and added these to the manuscript (SI Fig. 12).

5. The observation of one polymorph by cryoEM does not mean it is the only one or the most stable possible. The statement is still incorrect. There are many amyloid structures found as unique structures in their corresponding diseases, and yet, they are not particularly more stable than the same protein forming other structures. Please change.

Response: If there is essentially a single fibril structure in a patient, there must be a reason for this. Our explanation for this observation is a higher biological stability of the observed structure compared to other possible states. Biological stability means that a state can persist in a biological environment, which can arise from thermodynamic stability. However, biological stability arises primarily from a resistance to removal by proteases or macrophages, which does not necessarily arise from thermodynamics. We assume that the referee applies the term stability only to thermodynamic stability (ΔG values) and that this causes the confusion.

6. Protein aggregation starts by definition from the native state (whatever that is, unfolded, folded, semi-folded...). What is the alternative? where is the misfolding start if not the native fold? I see it important to highlight that the disulfide bonds remain in the amyloid structure. That is an interesting observation perhaps suggesting that the unfolding needs to happen, there is no reduction during the process.

Response: We are not aware that it would be part of any generally accepted definition that protein aggregation starts from a native state. Instead, it is often described that protein aggregates may be formed from polypeptide chains that have not reached the native state and became misfolded prior to adopting a native state. In addition, aggregates and amyloid fibrils can form from polypeptide chains that are intrinsically unable to fold into native or globular protein states, such as artificial polypeptide chains, polyamino acids or peptide fragments.

7. Did the author not discard any particles during 2D and 3D classification? This seems very unlikely (and not recommended). They also mentioned that "There was an essentially monomorphic distribution of fibrils with no discernible polymorphism", suggesting the presence of potential additional morphologies that were not discernible (therefore discarded?). Can the authors share all the 2D classes?

Response: We attempted to remove classes in the reconstruction process but this resulted in 3D maps with a poorer resolution. Therefore, we have not done it in the presented 3D map. Removal of classes generally bears the risk of introducing a data bias or of reducing the data quality. Below is a figure showing all classes obtained after 20 rounds of 2D classification. The particles became distributed across 94 classes, 6 classes were empty after 20 rounds.

8. The sentence "there is evidence from different amyloid diseases and from animals and humans that the fibril structure is consistent at different deposition sites of the same patient or animal, all suggesting that the observed fibril structure is representative for the fibrils in this patient" is incorrect (it also has a typo and lack references). There are several studies (including some of the authors) on

ATTR showing that patients with the same mutation (V30M) can show different structures depending on deposition site (vitreous humor vs heart). It is unclear whether the deposition does not vary the structure.

Response: The mentioned publications on V30M ATTR amyloidosis describe fibril structures from two different patients. Therefore, the observed structural differences may have arisen from different tissues (heart was analysed in the first study and eye in the second one) or from patient-specific differences. Patient-specific variations were recently reported for another mutation (I84S) in ATTR amyloidosis as well (Nguyen et al., 2024, PMID: 38233397). By contrast, when studies compared the fibril from different deposition sites of the same patient or animal, typically report consistent fibril structures at these different sites. Such evidence has now been reported for different forms of systemic amyloidosis (AA, AL, ATTR), for humans and for animals, for different methods of analysis and for data obtained in different laboratories, including the laboratories of Stefano Ricagno (Puri et al., 2023, PMID: 37516426) and Lorena Saelices Gomez (Nguyen et al., 2024, PMID: 38798361), and ours (Annamalai et al., 2017, PMID: 28544119).

9. What do the authors mean by "These findings are consistent with observations in other forms of amyloidosis, where ex vivo fibrils were found to be structurally different and more proteolytically stable than in vitro formed fibrils, and hence with the idea that pathogenically relevant amyloid fibril structures arose from a proteolytic selection mechanism"?

Response: We agree to the referee that the logic behind this part was not clear enough and partly redundant to an earlier paragraph. We rephrased it as follows: "These conclusions are consistent with observations that amyloidosis is associated with fibril morphologies of high proteolytic stability and the idea of pathogenic amyloid fibrils being proteolytically selected."

10. What do the authors mean by "We conclude that all previously described fibril proteins may be difficult to reconcile with the currently described fibrils structure and that the fibrils in these patients could be structured differently"?

Response: This conclusion was removed due to the new data obtained as described in comment 2.

Reviewer #2 (Remarks to the Author):

The authors have satisfactorily addressed the issues raised by this reviewer

Response: We thank this referee for the pleasant interactions and constructive remarks.

Reviewer #3 (Remarks to the Author):

The authors describe high resolution cryo-EM structure of patient-derived amyloid structure of mutant (D87G) lysozyme. The authors indicate that this is the first structure of such a fibril. Interestingly, the entire protein is well-resolved in the fibril state without the usual regions of unresolved disorder. As the first atomistically resolved structure of the class of lysozyme mutant fibrils, this is a noteworthy study. However, there are key points which I feel the authors should emphasize/expand and others in which the authors should de-emphasize/contract:

1. Is the disease due to loss of function of natively folded lysozyme or gain of function toxicity of the fibril form? The best consensus answer to this question is important to include because it determines the significance of the detailed structure of the fibril, which is the contribution of this work. If it is the former, then the shape of the fibril is only relevant so far as it informs on how much fibril can be formed (i.e. thermodynamic stability, kinetic accessibility, and proteolytic protection). If the answer is the latter, then knowledge of the shape of the fibril is important in its own right because of how it could inform on

the toxicity mechanism. This is crucial context for the findings of this work.

Response: We gratefully acknowledge the constructive criticism raised by this referee which helped us to further strengthen our manuscript. The patient's symptoms are rather consistent with a gain of function disease. The native function of lysozyme arises from its anti-bacterial activity so that a loss of function disease would imply the involvement of a bacterial infection. However, this is not observed, and the patient shows kidney dysfunction as well as amyloid deposition in the kidneys. Taken together, these findings argue for a gain of function mechanism.

2. As motivation for this work, the authors state that "In absence of detailed information about the three-dimensional (3D) structure of the formed fibrils it is difficult to further examine this possibility." I think this is too brief and indirect. The authors should instead state what types of structural characterization exists for lysozyme fibrils, and then state that lysozyme fibrils have not yet been characterized with atomic resolution. For example, kinetic and low-resolution structural characterization have been carried out by Ziaunys et al, Protein Science 2023, as well as by Matsuo et al, Frontiers in Molecular Biosciences, 2022.

Response: We thank this referee for this suggestion. While we acknowledge the importance of these previous studies we meant with the cited sentence that there are so far no PDB-files of patient-derived lysozyme fibrils. We have now clarified this in the revised version of the manuscript: "In absence of coordinate files showing the three-dimensional (3D) structures of the patient-derived ALys fibrils it is difficult to further examine this possibility". The publications suggested by the referee actually investigated in vitro fibril formation of hen egg white lysozyme - not human lysozyme. Therefore, it is difficult from these studies to extrapolate to the effects of mutations on the human protein.

3. Molecular dynamics simulations.

3a. It was never stated how many copies of the monomer were simulated. From Fig. 3b it looks like 5 or 6 monomer layers are included in each simulation. Is that correct? If so, the thinness of the layers would introduce bending and twisting modes of the systems that would not be present in experimentally observed fibrils. These deformations are incompatible with maintaining the in-register monomer-monomer stacking, leading to effective shear forces that could decouple the end-monomers. Even for the 87G mutant fibril, it looks like there are conformational fluctuations and decoupling of the end-monomers in Fig. 3b. Therefore, this simulation setup potentially introduces artifacts that artificially destabilize the fibril. I agree with the authors that the reversion of the mutant G87 to D is likely to destabilize the fibril due to negative charge repulsion and steric clashes, but a devil's advocate could say that the unfolding observed in the reverted simulations in Fig.3b is due to magnification of the destabilization by this simulation artifact. It could be argued that the D87 could in fact locally deform the fibril to make more space for the larger side chain without significantly altering/unfolding the fibril fold. The authors stated that they used a cubic box. In that case, this issue could be avoided by repeating the simulations in the same sized-box but with many more monomer copies (e.g. 20 or 30).

Response:

We thank the referee for the comment. We previously analyzed the stability of fibrils with atomistic and coarse-grained simulations and showed that fibrils with more than 4 peptides form a stable cross- β structure (Schwierz et al., 2015, PMID: 26694883, Supporting Information Figure 1). The fibrils with 6 peptides used here therefore serve as ideal minimal model system. The reason for using such minimal model systems in our all-atom simulations is that larger numbers of monomers require significantly larger simulation boxes to avoid interactions with the periodic images. For instance, a fibril with 20 monomers would require a box length of about 20 nm corresponding to about N=1 Mio atoms (compared to 0.4 Mio atoms for 6 monomers). Even though such simulations are feasible on supercomputers, they would require enormous computational resources due to the scaling of computing time with NlnN. For the minimal system, we carefully checked for bending and twisting motions in the WT and patient sequence simulations and confirmed that there is no indication of such artefacts. This is shown in the movies provided here:

Conformational dynamics of the PT-sequence (links can be opened within the Chrome browser):

https://td-host.rz.unibw-muenchen.de/ccgkxdt/public/5etR2spp?k=LJdfTJy8vt8k-HWWP-mDH0AGlnS9RR6RMR3yO_HXMFk

Conformational dynamics of the WT-sequence:

https://td-host.rz.unibw-muenchen.de/ccgkxdt/public/~xPVSg7P?k=gdbhTa1aepWpM3Wlo_Ysmiz2k0aF_pLqnwruq4c3uf8

We included these important points in the method part of the revised manuscript (pages 17-19).

3b. In methods, authors stated that they used the PDB Manipulator tool to replace G87 by D and to position the D side chain. What were the constraints of this process? For example, was positioning done to optimize inter-molecular side-chain-packing with other monomers? Was the threshold for the following energy minimization? Details of this process should be provided. The starting point of the MD simulations for the homology-modeled WT fibril could be sensitive to the choices made in this process. This would then affect the stability of the system in the subsequent MD simulations. It is possible that there exists lower-energy alternative packings of the D87 that the MD does not sample in the relatively short time of 200 ns.

Response: We thank the referee for the question. The algorithms of the PDB Manipulator tool replaces G87 with D87 such that the backbone dihedrals remain unchanged. In addition, the NH₂ atoms, which are identical in both residues, are placed in the same position (see Figure 1). Due to these constraints, there is no alternative starting configuration for the homology-modeled fibril. In the subsequent energy minimization, we use the steepest descent algorithm to obtain the minimum energy configuration. The minimization procedure converged after 50,000 steps (see also Methods part of the revised manuscript).

Figure 1: Superposition of the cryo-EM structure of the patient sequence and the G87D mutation modelled with the CHARMM-GUI. D87 is shown in blue. The VdW radius of the G87 atoms are indicated by spheres since the atoms of the backbone and the NH₂ group are on top of the corresponding atoms in D87. The two neighboring amino acids are shown in licorice representation. The backbone dihedrals are not affected by the mutation procedure.

3c. It is standard practice to run 3 independent MD simulations at each condition to demonstrate that the results are robust to sampling variations arising from the stochastic nature of molecular dynamics.

Response: We agree with the referee. To improve the sampling, we performed additional simulations for both systems. In total, we performed 6 x 200 ns of production runs for the patient and WT sequence (see Method section in the revised manuscript). In total, total 2.4 μ s of simulation time were used. The maximum distance between the residues in the terminal chains shows that there is a dissociation of the molecules at one end of the fibril close to position 87D of the WT sequence (see Fig. SI 13 in the Supporting Information).